# Association of parental characteristics and emotion regulation in children and adolescents with and without psychopathology: A case-control study

Eva-Maria Fassot[1]*, Brunna Tuschen-Caffier[1], Julia Asbrand[1,2]

**1** Department of Clinical Psychology and Psychotherapy, Institute for Psychology, University of Freiburg, Freiburg, Germany, **2** Department of Psychology, Faculty of Life Sciences, Humboldt University of Berlin, Berlin, Germany

☯ These authors contributed equally to this work.
* eva-maria.fassot@psychologie.uni-freiburg.de

**Data Availability Statement:** Data cannot be shared publicly as this is not included in the informed consent by participants and the mental health data of children is particularly sensitive. To

## Abstract

This study explores the difference in child emotion regulation (ER) and parenting between a heterogeneous clinical sample (ClinS) and a community sample (ComS). We hypothesized that parents of the ClinS would report more dysfunctional child ER and more dysfunctional parenting regarding the child's negative emotions than parents of the ComS. Further, we aimed to predict child ER by parenting behavior, parents' ER, and mental health. Parents of children and adolescents (aged 6–18 years) seeking treatment at an outpatient clinic were compared to a matched sample of parents in a ComS (n = 57 each group). As predicted, the children in the clinical group were reported to use less reappraisal and more suppression than ComS children. No difference was found in dysfunctional emotion parenting between the groups. Reappraisal in parents and supportive reactions to negative emotions predicted reappraisal in children. No predictor was found for child suppression. Child emotion regulation and parents' psychopathology were not associated. These results could suggest new elements for prevention and intervention programs with parents concerning their own emotion regulation and their reaction to negative emotions in children.

## Introduction

Emotion regulation (ER) and its socialization are critical elements in the development of children and have become a primary focus of research in recent decades (e.g., [1, 2])). Thompson provided a general definition of ER as "extrinsic and intrinsic processes responsible for monitoring, evaluating and modifying emotional reactions, especially their insensitivity and temporal features, to accomplish one's goal" (pp. 27–28) [3]. ER has been previously linked to psychopathological symptoms also in children [4–6] and is discussed to be an underlying transdiagnostic factor for psychopathology [6]. In this study, we try to identify parental characteristics that are associated with the ER in healthy children and children with psychopathology.

ask them to publicize the data could disturb the confidential relationship with their therapist. However, deidentified participant data with annotations will be made available to other researchers upon reasonable request (towards the study administration at kinderprojekt@psychologie. uni-freiburg.de.)

**Funding:** The article processing charge was funded by the Baden-Wuerttemberg Ministry of Science, Research and Art and the University of Freiburg in the funding programme Open Access Publishing. The funders had no role in the study design or data collection.

**Competing interests:** I have read the journal's policy and the authors of this manuscript have the following competing interest: The authors declare they have no financial interests. Prof. Tuschen-Caffier is the head of the institute the data was collected. This does not alter our adherence to PLOS ONE policies on sharing data or materials.

From a clinical perspective, it could be helpful to identify parental characteristics that are associated to ER in children to improve therapeutic interventions.

In a process model of ER, Gross [7] defined strategies that are used before the emotional reaction (i.e., antecedent strategies) and after (i.e., response-focused strategies). antecedent strategies are believed to be more effective than response-focused strategies because the emotional response is inhibited before its complete activation, which changes one's behavior. In contrast, response-focused strategies are believed to be less effective because the influence of the ER is limited [7].

Two widely explored ER strategies are reappraisal and suppression (e.g., [8, 9]). Reappraisal describes the attempt to reinterpret a distressful situation by changing thoughts and beliefs about it and is classified as an antecedent strategy [8, 10]. There is evidence that habitual reappraisers benefit from using reappraisal as it decreases the experience of negative emotions without cognitive or social costs [10]. Suppression is defined as the conscious inhibition of ongoing emotion-expressive behavior and is a response-focused strategy and it is categorized as a maladaptive strategy [7]. The regular use of suppression is associated with more experience of negative emotion and less experience of positive emotion [8].

Investigations on the development of ER have focused predominately on early childhood and infancy, as significant changes in emotional understanding and expressing emotion occur in this period (e.g., [11]) During middle childhood the variety of ER strategies expands [3]. For example, children at this age understand that emotional experiences can be changed by internal emotional redirecting or by external distraction (e.g., [12]). Preschool-age children already use both reappraisal and suppression strategies (e.g., [5, 9, 13]). Research on normative age-related emotion regulation patterns during middle childhood and adolescence is inconsistent [13–15].

Predominantly the two strategies are linked to psychopathology. For example, Aldao et al. [16] showed in their meta-analysis a positive correlation between suppression and anxiety, depression, and substance abuse and a negative correlation between reappraisal and anxiety and depression [16].

Similar to findings in adults, there is evidence of a relation between reappraisal, suppression, and psychopathology in children and adolescents [5, 17, 18]. Less use of reappraisal and more use of suppression was found to be associated with well-being and healthier personal interactions in adolescents [8]. Further, there is evidence of an association between the two strategies and psychopathology like anxiety and depression in children and adolescents [4, 5, 19]. Shedding light on the direction of the relation, studies have shown that emotional dysregulation predicts psychopathology [20–22]. Thus, ER is a possible risk factor for the development of psychopathology in children and adults. Still, precursors of ER should be examined more closely.

There is an increasing body of research on the socialization of emotion and the influence of especially parents on the ER and the psychopathology of their children. Several parental characteristics have been suggested to have an influence. Morris et al. offered a theoretical framework for socializers' impact on ER [23]. They proposed a tripartite model of family impact on children's ER consisting of three components of emotional socialization of parents—the observation, the parenting practice, and the emotional climate of the family—that directly influence the ER of children. Further, they proposed that parental characteristics such as psychopathology influence the ER of children in an indirect way.

Informed by the tripartite model, we aimed to shed light on factors that may influence suppression and reappraisal in children and their psychopathology. One component of parenting behavior is the reaction of parents to negative emotions in their children [23]. Researchers have divided this reaction into supportive (i.e., emotion- or problem-focused behavior,

expressive encouragement) and unsupportive (i.e., minimization or punitive reaction or distress responses) reactions to negative emotions (e.g., [24, 25]). There is evidence that supportive reactions help children regulate their behaviour and increase social functioning and coping, whereas unsupportive reactions are related to lower levels of social competence [25, 26].

While several studies have targeted healthy children, research on children with psychopathology remains scarce. Only a link between anxiety and unsupportive reactions to negative emotions was found for children aged 7 to 13 years [27]. Thus, one of our aims was to assess the association between psychopathology in children and the reactions of parents to their negative emotions. Further, research has so far largely overlooked the relation between parental reaction to children's emotions and children's reappraisal and suppression. There is evidence of a relation between adolescents' use of reappraisal and suppression and their retrospective ratings of parental care or unsupportive parenting behavior that even predicted trait anxiety (e.g., [28, 29]). Thus, there is preliminary evidence of a link between socialization of ER, social competence, and psychopathology in children that even influences psychopathology in adulthood. In the present study, we investigated this relation in actual perception and not in retrospective reports.

In addition to parenting behavior, parents' own ER has been thought to have an impact on ER and psychopathology in children [23, 30] There is some evidence of a positive correlation between parents' and children's suppression [9, 14]. However, the direction of the correlation between ER in parents and reappraisal in children remains unclear and has not been fully investigated. Still, parental use of suppression was associated with less use of reappraisal in children; that is, suppression in parents might inhibit the use of more adaptive strategies [9]. Further, there is some evidence that parents' ER is associated with their reaction to negative emotions, problems in ER in parents lead to unsupportive behavior [31]. In this study, in addition to the reaction to negative emotions by parents, we investigated the relation between reappraisal and suppression in parents and reappraisal and suppression in children.

There is also evidence of an association between parents' psychopathology and problems in parenting. The influence of parents' psychopathology on ER in children, which has received little attention to date, was thus also considered in this study. In the tripartite model, Morris et al. proposed a direct influence of parents' psychopathology on parents' teaching style and an indirect influence on children's ER [23]. When parents have psychopathology symptoms and problems regulating their own emotions, it might be harder for them to teach or support their children, especially when they are in a distressful situation. There is evidence that parental psychopathology influences their parenting style and the behavioral problems of their children [32–35]. Mothers with psychopathology symptoms showed less supportive reactions to negative emotions and tended to ignore negative emotions of their children (e.g., [36]) Much research has been conducted on mothers with depression, their deficits in parenting, and the mental health of their children [37, 38]. There is also some research concerning the link between depression in mothers and problems in the ER of their children [39]. Taken together, these results suggest that parental psychopathology influences their parenting behavior, their own ER, and the mental health of their children. But this association has not been investigated and is, thus, the aim of this study.

## The current study

The present study investigated the differences in ER and parenting behavior between a sample with a vast range of age and heterogenous psychopathology and a community sample. Further with the tripartite model as a theoretical foundation, we investigated the relation between (a)

the ER of parents, (b) one aspect of parenting behavior, namely, the reaction to negative emotions, and (c) reappraisal and suppression in children, as reported by the parents. Furthermore, we explored the indirect influence of parents' psychopathology on ER in children. Previous studies investigated the factors separately. We wanted to investigate the influence of these factors together on two specific strategies: reappraisal and suppression.

First, we collected data from children with a mental disorder (ClinS) and children from a community sample (ComS) to compare ER between the two groups. We expected that children in the ClinS would use the adaptive ER strategy reappraisal less often and the maladaptive ER strategy suppression more often than children in the ComS (Hypothesis 1) We further hypothesized that parents of children with a mental disorder would report unsupportive reactions to negative emotions more often and supportive reactions to negative emotions less often than parents of the ComS children (Hypothesis 2). Taking Morris et al.'s tripartite model [23] and current research into account, we hypothesized that the factors parents' psychopathology, lower reappraisal, higher suppression, and maladaptive reaction to their child's negative emotions would predict the ER of their child. The direction of the effect was expected to be positive for the child's suppression and negative for the child's reappraisal (Hypothesis 3). As gender and age of the children might influence ER and parenting behavior, we calculated the different analyses with gender and age as supplemental variables in an explanatory fashion and reported this in S1 File. Regarding gender differences and age no specific hypotheses were made because research concerning age and gender is inconsistent [13–15].

## Method

### Participants

The sample comprised $N = 229$ parents (202 mothers and 27 fathers) of children aged 6 to 18 years. Parents completed several self-report instruments. We obtained written informed consent from parents and children older than 11 years. Younger children (<11 years) were informed orally. The local ethic committee of the Albert-Ludwigs-University, Freiburg approved the study (Ethik-Kommission Freiburg: date of approval: 03.22.2016, approval number: 66/16).

**Clinical Sample (ClinS).** A state-approved institute for psychotherapy for children associated with a German university recruited $n = 106$ patients ($M_{age} = 14.09$ years, $SD = 2.44$) and their parents. The children, later on, received cognitive behavioral therapy for different diagnoses in an outpatient setting (see Table 1). Comorbidity was allowed and was present in 31 (29%) of patients. Diagnoses were verified by a structured clinical interview conducted with the parents (Diagnostisches Interview bei Störungen im Kindes- und Jugendalter; [40]). The interview has proven to be a reliable and valid instrument [41].

**Control group: Community Sample of families (ComS).** A community sample (ComS) of families was recruited to participate in an online survey through the distribution of flyers in schools mostly in southwest Germany as well as flyers displayed in medical offices, local sport clubs, or different online forums. Participants were further encouraged to participate by being offered a lottery for a 20-euro voucher. One hundred twenty-three parents and children ($M_{age} = 11.28$ years, $SD = 3.21$) finished the survey and could be included. A comparison of sociodemographic data of the groups can be found in Table 2. Groups differed in the parent's education level and income level.

### Material

**Sociodemographic data.** In the community sample, sociodemographic data such as the age, profession, and income of the parents and the age and type of school of the children were

**Table 1. Frequency and percentage of primary diagnoses in the clinical sample (*n* = 106).**

| Primary Diagnose (ICD 10) | | Frequency | Percentage |
|---|---|---|---|
| | | *n* | % |
| F20-29 | Schizophrenia, schizotypal and delusional disorders | 1 | 0,9 |
| F30-39 | Mood [affective] disorders | 19 | 18 |
| F40-49 | Neurotic, stress-related, and somatoform disorders | 32 | 30 |
| F50-59 | Behavioural syndromes associated with physiological disturbances and physical factors | 4 | 4 |
| F90-99 | Behavioural and emotional disorders with onset usually occurring in childhood and adolescence | 38 | 36 |
| **Missing** | | 11 | 10 |

*Note*. Percentages do not always add up to 100 because of rounding.

collected in a separate questionnaire at the beginning of the online survey. In the clinical sample, data were collected from the standard documentation of the outpatient clinic.

**Psychopathology.** *Parents*. Parents self-reported symptoms of psychopathology by completing the Symptom Checklist 27 (SCL-27; [42]) a shortened version of the Symptom Checklist 90 ([43]; German version [44]). To provide a more holistic picture, 13 items from the Brief Symptom Inventory (BSI; [45]) were added (Obsession-Compulsion, Anxiety, and Hostility

**Table 2. Sociodemographic data of the parents of the Clinical Sample (ClinS) and the Community Sample (ComS).**

| Variable | Clinical Sample (N = 106) | | Community sample (N = 123) | | Statistics |
|---|---|---|---|---|---|
| | *n* | % | *n* | % | $X^2$ (*df* = 1) |
| **Person who completed the questionnaire** | | | | | |
| Mother | 98 | 92 | 104 | 85 | 3.41; n.s. |
| Father | 8 | 8 | 19 | 15 | |
| **Mother's highest education level** | | | | 19 | 31.15** |
| Bachelor's degree or higher | | 23 | 45 | 37 | |
| Less than bachelor's degree | 54 | 51 | 77 | 63 | |
| Missing | 28 | 26 | 1 | >1 | |
| **Father's highest education level** | | | | | 42.27** |
| Bachelor's degree or higher | 33 | 31 | 56 | 46 | |
| Less than bachelor's degree | 44 | 42 | 63 | 51 | |
| Missing | 29 | 27 | 4 | 3 | |
| **Income level (monthly, net in euros)** | | | | | 17.02** |
| <1,300 | 4 | 4 | 3 | 2 | |
| 1,301–2,500 | 35 | 33 | 27 | 22 | |
| 2,501–3,500 | 20 | 19 | 42 | 34 | |
| 3,501–5,000 | 17 | 16 | 27 | 22 | |
| >5,000 | 4 | 4 | 24 | 20 | |
| Missing | 26 | 25 | 0 | 0 | |
| **Child's gender** | | | | | 2.29; n.s. |
| Female | 58 | 55 | 55 | 45 | |
| Male | 48 | 45 | 68 | 55 | |

*Note*. Percentages do not always add up to 100 because of rounding.

*$p < .05$

**$p < .01$

n.s: not significant.

scales). The complete checklist consisted of 40 items. An overall sum score the Global Severity Index (GSI) of the six scales of the SCL-27 and the three added scales of the BSI, consisting of 40 items was used as the independent variable "mental health" to test Hypothesis 3. The internal consistency of this score was $\alpha = .93$.

*Children*. Parents in the clinical sample completed the German version of the Child Behavior Checklist (CBCL/4-18; [46]; original: [47]) or the revised version of the Child Behavior Checklist (CBCL/6-18R; [48]), because the revised version was added later to the survey for the ClinS. Parents of children in the ComS completed only the revised version (CBCL/6-18R). The CBCL is an instrument to screen for emotional and behavioral problems in children and adolescents. The 118 or 120 items (depending on the version) address a wide range of behavioral and emotional problems observed in the last 6 months. In the analysis, eight syndrome scales (Anxious/Depressed, Attention Problems, Rule-Breaking Behavior, Withdrawn/Depressed, Somatic Complaints, Social Problems, Thought Problems, Aggressive Behavior) and two higher scales (Internalizing and Externalizing Problems) and a total problem score can be assessed. The total problem score was used in the current study. The CBCL was used as a screening instrument to ensure that no participants with a clinically noticeable disorder participated in the community sample; that is, children in the community sample with a total problem score higher than 70 were excluded before the analysis [49].

**ER strategies.** Parents completed the Emotion Regulation Questionnaire (ERQ; [50]; German version: [51]) to measure the ER strategies expressive suppression and cognitive reappraisal. The German version has been reported to show good internal consistency [51]. In the current sample, the internal consistency was good: $\alpha_{Reappraisal} = .85$, $\alpha_{Suppression} = .70$. Parents further rated their child's ER strategies in a previously validated ERQ version for children [52] Internal consistency for the Suppression scale was $\alpha = .66$, and for the Reappraisal scale, $\alpha = .86$.

**Reaction of parents to the emotions of their children.** To measure parents' reactions to negative emotions of their children, parents completed the Coping with Children's Negative Emotions Scale (CCNES; [53] or the Coping with Children's Negative Emotions Scale—Adolescent Perception version (CCNES-AP; [54]). The CCNES-AP is for parents of children older than 11 years and the CCNES is the version for children younger than 11 years old. The CCNES consists of situations in which children may experience negative emotions. In the CCNES, parents are asked to identify how they would respond to 12 different scenarios; in the CCNES-AP they are asked about their reaction to nine scenarios. Each scenario has six responses that parents rate on a 7-point Likert scale regarding their likelihood of responding that way (1 = *very unlikely*, 7 = *very likely*). This questionnaire has six subscales: Emotion Focused, Problem Focused, Minimization, Punitive, Expressive Encouragement, and Distress Responses. A factor analysis by Fabes et al. [53] revealed that there were two factors, supportive and unsupportive reaction to negative emotions, which were created from only four subscales. As recommended by Gunzenhauser et al. [9] for the German version the supportive reaction factor consists of the subscales Emotion Focused and Problem Focused, and the unsupportive reaction factor consists of the subscales Minimization and Punitive. These subscales were used to operationalize the hypotheses 2 and 3. For this study, two bilingual psychologists translated and re-checked the questionnaire. The internal consistency for CCNES supportive reaction was $\alpha = .65$ and for unsupportive reaction, $\alpha = .75$. The internal consistency for CCNES-AP supportive reaction was $\alpha =. 79$ and for unsupportive reaction, $\alpha = .53$.

## Procedure

In the clinical sample, the material was integrated into the normal diagnostic process at the beginning of treatment. Participation was voluntary and preceded by written informed

consent. In the community sample, parents participated in an online survey, which took about 45 min. At the beginning of the study, they gave written informed consent. The data were saved anonymously. In the end, participants had the opportunity to give their email separately to participate in the raffle for the voucher.

## Data analysis

All statistical analyses were calculated using IBM SPSS version 25. The difference in age of the children between the two groups was significant, $t_{(227)} = -7.37$, $p = .002$. Also, the income level and the education level of the parents differed significantly (see Table 2). To achieve comparability between the two groups, case-control matching was performed with the matching factors age and sex, because these factors are known to influence the ER and parenting behavior [13–15]. We decided not to match more variables to keep an acceptable sample size. A comparison of sociodemographic data of the matched groups can be found in Table 3. Case-control matching is an iterative process that led to a sample of $n = 57$ cases in each group. After the matching process, the age of the children was $M = 13.68$ years ($SD = 2.58$) in both groups. To test the first and second hypotheses, the matched sample was used. Because the ERQ for parents was added later to the survey, the third hypothesis was tested within another subsample ($n = 139$).

**Table 3. Sociodemographic data of the parents of the matched Clinical Sample (ClinS) and the Community Sample (ComS).**

| Variable | Clinical Sample (N = 57) | | Community sample (N = 57) | | Statistics |
|---|---|---|---|---|---|
| | *n* | % | *n* | % | $X^2$ (*df* = 1) |
| **Person who completed the questionnaire** | | | | | |
| Mother | 54 | 95 | 49 | 86 | 2.52; n.s. |
| Father | 3 | 5 | 8 | 40 | |
| **Mother's highest education level** | | | | | 19.95** |
| Bachelor's degree or higher | 17 | 30 | 23 | 40 | |
| Less than bachelor's degree | 24 | 42 | 34 | 60 | |
| Missing | 16 | 28 | 0 | 0 | |
| **Father's highest education level** | | | | | 19.41* |
| Bachelor's degree or higher | 18 | 31 | 29 | 51 | |
| Less than bachelor's degree | 24 | 42 | 28 | 29 | |
| Missing | 15 | 26 | 0 | 0 | |
| **Income level (monthly, net in euros)** | | | | | 15.25** |
| <1,300 | 3 | 5 | 1 | 1 | |
| 1,301–2,500 | 20 | 35 | 27 | 22 | |
| 2,501–3,500 | 9 | 16 | 24 | 42 | |
| 3,501–5,000 | 9 | 16 | 12 | 21 | |
| >5,000 | 2 | 4 | 10 | 18 | |
| Missing | 14 | 24 | 0 | 0 | |
| **Child's gender** | | | | | |
| Female | 28 | 49 | 28 | 49 | |
| Male | 29 | 51 | 29 | 51 | |

*Note.* Percentages do not always add up to 100 because of rounding.

*$p < .05$

**$p < .01$, n.s: not significant.

To test the first hypothesis, we calculated a repeated-measures analysis of variance (MAN-OVA) with the within-subject factors group and ER strategy (dependent variables: Suppression and Reappraisal scales of the parent reported ERQ for children). In the case of significance, we calculated post hoc two *t-tests* for dependent samples. For the second hypothesis, we again used a repeated measures MANOVA with the within-subject factors group and parenting behavior (dependent variables: z-standardized CCNES, supportive and unsupportive scale scores of the two questionnaires). To test the third hypothesis, two hierarchical multiple regressions were conducted to predict either suppression or reappraisal in children. In a first step, the reaction of the parents to negative emotions in their children (*z*-standardized CCNES, supportive and unsupportive scale scores) were included. In a second step, parents' ER scores (z-standardized ERQ reappraisal, z-standardized ERQ suppression) were included. Finally, parents' psychopathology was included in a third step (z-standardized GSI)

## Results

### Mental disorders and ER

The repeated measures MANOVA revealed a significant effect of ER strategy, $F(1, 56) = 264.48$, $p < .001$, partial $\eta^2 = .83$, but no effect of group, $F(1, 56) = .03$, $p = .857$, partial $\eta^2 = .01$, and a significant interaction effect of Group × Strategy, $F(1, 56) = 17.58$, $p < .001$, partial $\eta^2 = .24$. Children in the ComS were reported by their parents to use the ER reappraisal strategy more often than children in the ClinS, $t(56) = -2.39$, $p = .020$. Children in the ClinS were reported by their parents to use suppression more often than children in the ComS, $t(56) = 2.79$, $p = .007$. Means and standard deviations of the groups are shown in Table 4.

### Reaction of parents to negative emotions of their children

To test the difference in parents' reactions to their child's negative emotions in the ClinS and the ComS, a repeated measures MANOVA with the factors group and parenting behavior was performed. There was no significant effect of parenting behavior, $F(1, 56) = 0.11$, $p = .747$, $\eta^2 = .002$, or of group, $F(1, 56) = 0.86$, $p = .359$, $\eta^2 = .015$, and no interaction effect of Group × Parenting Behavior, $F(1, 56) = .299$, $p = .587$, $\eta^2 = .005$. The z-standardized means and standard deviations are shown in Table 5. There was no difference between the parents of the ClinS and those of the ComS in using a supportive or unsupportive reaction to negative emotions of their children.

### Prediction of children's ER

**Reappraisal.** To test the third hypothesis, 139 cases could be included, in two separate hierarchical multiple regressions (reappraisal and suppression, respectively; only 139 cases could be reported because the ERQ for parents was added later to the survey). As there were significant deviations from normality in the variable mental health ($z_{kurtosis} = 12,18$; $p < 0.001$; $z_{skewness} = 11,75$, $p < 0.001$), a bootstrapping procedure using 1,000 samples was used in the hierarchical multiple regressions. As shown in Table 6 for reappraisal, the first model step

**Table 4. Means and standard deviations of reappraisal and suppression in the clinical sample and the community sample for the matched sample.**

| Sample | N | Reappraisal | | Suppression | |
|---|---|---|---|---|---|
| | | *M* | *SD* | *M* | *SD* |
| Clinical | 57 | 23.12 | 7.65 | 12.42 | 6.00 |
| Community | 57 | 26.20 | 7.13 | 9.64 | 4.45 |

**Table 5. Means and standard deviations (z-Standardized) of supportive and unsupportive reactions of parents in the clinical sample and the community sample for the matched sample.**

| Sample | N | Supportive | | Unsupportive | |
|---|---|---|---|---|---|
| | | *M* | *SD* | *M* | *SD* |
| Clinical | 57 | 0.08 | 1.04 | 0.06 | 0.92 |
| Community | 57 | -0.12 | 1.05 | -0.01 | 1.05 |

including parents' reaction to their child's negative emotions was significant, $F(2, 136) = 5.56$, $p = .005$, adjusted $R^2 = .062$. The second model step including parents' ER was also significant, $F(4,134) = 4.11$, $p = .004$. The third step including of parents' mental health remained significant, $F(5, 133) = 3.29$, $p = .008$. However, no additional variance could be explained (see Table 6). Table 6 shows the standardized and unstandardized regression coefficients with bootstrapped confidence intervals.

**Suppression.** A second hierarchical multiple regression was calculated to predict suppression based on parents' reactions to negative emotions of their children at Step 1, emotion regulation in parents at Step 2, and mental health at Step 3. Once again, a bootstrapping procedure based on 1,000 samples was applied. No significant effects were found for the model. The first model was not significant, $F(2, 136) = 0.06$, $p = .943$, $R^2 = .001$, adjusted $R^2 = -.014$. Including emotion regulation strategies of parents did not significantly change the explained variance, $F(4, 134) = 0.38$, $p = .823$. Finally, including mental health of parents did not increase the explained variance, $F(5, 133) = 0.30$, $p = .910$. Table 7 shows that our model could not predict suppression.

## Discussion

This study investigated the potential influence of parental characteristics on parent reported children's ER reappraisal and suppression strategies in a clinical sample (ClinS, i.e., children with a mental disorder) versus a community sample (ComS; [23]). Children in the CS used more reappraisal and less suppression than children in the clinical sample. There was no

**Table 6. Unstandardized *(b)* and standardized (β) regression coefficients for each predictor in a hierarchical regression model predicting reappraisal in children.**

| Predictor | b | 95% CI for b | | SE β | β | $R^2$ | $\Delta R^2$ | p |
|---|---|---|---|---|---|---|---|---|
| | | LL | UL | | | | | |
| Step 1 | | | | | | .77 | .08 | .005 |
| Constant | 22.18 | 20.94 | 23.34 | .65 | | | | ≤.001 |
| z-CCNES unsupportive | 0.54 | -0.90 | 1.91 | .68 | .066 | | | .423 |
| z-CCNES supportive | 2.195 | 0.73 | 3.59 | .69 | .263 | | | .002 |
| Step 2 | | | | | | .11 | .03 | .081 |
| Constant | 22.19 | 20.95 | 23.33 | .64 | | | | ≤.001 |
| z-ERQ reappraisal parent | 1.49 | 0.20 | 2.73 | .67 | .189 | | | .027 |
| z-ERQ suppression parent | 0.28 | -0.91 | 1.43 | .65 | .036 | | | .672 |
| Step 3 | | | | | | .11 | .001 | .761 |
| Constant | 22.12 | 20.75 | 23.66 | .68 | | | | ≤.001 |
| z-GSI mental health parent | 0.76 | -1.90 | 2.08 | .76 | -.026 | | | .761 |

*Note. N* = 139. CI = Confidence interval (based on 1,000 bootstrapped samples); *LL* = lower limit; *UL* = upper limit; z-CCNES = mean z-standardized score on the Coping with Children's Negative Emotions Scale; z-ERQ = mean z-standardized score on the Emotion Regulation Questionnaire; z-GSI = mean z-standardized score on the Global Severity Index.

**Table 7. Unstandardized *(b)* and standardized (β) regression coefficients for each predictor in a hierarchical regression model predicting suppression in children.**

| Predictor | *b* | CI | | SE β | β | $R^2$ | $\Delta R^2$ | *p* |
|---|---|---|---|---|---|---|---|---|
| | | *LL* | *UL* | | | | | |
| Step 1 | | | | | | .001 | .001 | .943 |
| Constant | 11.17 | 10.75 | 12.61 | .47 | | | | ≤.001 |
| *z*-CCNES unsupportive | -0.05 | -0.9 | 1.91 | .49 | -.017 | | | .847 |
| *z*-CCNES supportive | -0.136 | -1.11 | 0.79 | .505 | -.023 | | | .788 |
| Step 2 | | | | | | .011 | .01 | .823 |
| Constant | 11.66 | 10.77 | 12.58 | .477 | | | | ≤.001 |
| *z*-ERQ reappraisal parents | 0.02 | .-1.07 | 1.09 | .489 | .003 | | | .973 |
| *z*-ERQ suppression parents | 0.572 | -.319 | 1.45 | .483 | .105 | | | .238 |
| Step 3 | | | | | | .011 | .00 | .910 |
| Constant | 11.68 | 10.66 | 12.64 | .504 | | | | ≤.001 |
| *z*-GSI mental health parent | 0.52 | -1.23 | 1.13 | .589 | .008 | | | .928 |

*Note. N* = 139. CI = Confidence interval (based on 1,000 bootstrapped samples); *LL* = lower limit; *UL* = upper limit; z-CCNES = mean z-standardized score on the Coping with Children's Negative Emotions Scale; z-ERQ = mean z-standardized score on the Emotion Regulation Questionnaire; z-GSI = mean z-standardized score on the Global Severity Index.

difference in parental use of supportive or unsupportive reactions to negative emotions between the two groups. Finally, we also aimed to predict ER in children by different characteristics of parents. A supportive reaction to negative emotions and reappraisal in parents predicted reappraisal. Mental health in parents did not influence reappraisal in children. Parental characteristics did not predict suppression.

## Use of reappraisal and suppression in healthy children and children with mental disorders

Consistent with previous theoretical findings [17, 55] children in the ComS were reported to use more reappraisal by their parents than children in the ClinS. Our findings are in line with previous results of a negative relation between reappraisal and psychopathology for adults [16] that can be extended to children and adolescents. The earlier finding that reappraisal in adolescents and children was correlated with several positive consequences such as well-being and better interpersonal functioning in healthy children [56] can be supplemented with these new findings on a clinical sample of children and for a broad spectrum of ages. So far, there has only been evidence for samples with a smaller age range or a specific mental disorder such as anxiety or depressive symptoms [4, 17, 55].

Also in line with the literature [16, 18], there was a difference between the two groups for suppression. Parents reported that children in the ClinS used suppression more often than children in the ComS. Previous findings were mostly reported for adults [16] and can now be extended for children and adolescents. The previous findings for internalizing problems such as anxiety and depression [4, 17, 18] seem to be valid also for a larger variety of mental disorders, including externalizing problems as in our sample. The results support the transdiagnostic approach [6, 57, 58]. Problems in ER seem to be a general mechanism underlying mental disorders. It could be helpful to include the analysis of ER in the diagnostic process and to offer ER training even in programs for children with different mental disorders, as has been proposed, for example, by Heinrichs et al. [59]. The results did not change including age as a covariate and gender as an additional factor (see S1 File).

## Parenting and mental disorders in children

Contrary to our assumption, there was no difference in parenting behavior between the two groups. Parents with children in the ComS did not report more supportive parenting behavior or less unsupportive behavior in response to negative emotions than parents of children in the ClinS. Even the age or gender did not influence the reaction (see S1 File). Previously reported effects of the positive influence of supportive parenting behavior concerned social functioning and competence [25] and have not yet been shown for mental health. Maybe "mental disorder" as a category is too broad. There is evidence for differences in parenting behavior affecting anxiety disorders [27, 60] and depression in adolescents [61]. However, no difference in the use of supportive or unsupportive reactions between parents of children with attention-deficit disorders and parents of healthy children was found [62], which is in line with our results.

Possibly, assessment of parents' reaction to their child's negative emotions only falls short in the complex parent-child interaction and should be accomplished for positive emotions where maybe can be find a difference. Some studies have found differences in the reaction of parents to positive emotions in depressed versus healthy adolescents [61, 63]. Parents of depressed adolescents showed less acceptance of adolescents' positive affect and more often used strategies that dampened adolescents' positive affect than parents of healthy adolescents [61]. This should be investigated in other mental disorders. Focusing on and encouraging positive emotions could be a very important parental behavior, even more, important than parents' reaction to negative emotion. Other methodological considerations target the six subscales of the CCNES. Maybe these subscales do not display the entirety of all possible reactions to negative emotions. Mirabile [64] proposed "ignoring the child's emotion" as an additional and independent reaction to negative emotion. In that study, "ignoring" seemed to be an independent, reliable, and additional subscale. A negative relation between ignoring and general competence (e.g., resourceful, engagement in school) has also been found [65].

## Predictions of children's ER

This study increases the understanding of parents' emotion socialization behavior and its influence on parent reported reappraisal and suppression in children. The results suggest that a supportive reaction in parents was an important factor for the use of reappraisal in children. This is in line with the retrospective findings of Cabecinha-Alati et al. [28], who found the same association for adolescents and could be extended for adolescents and younger children in a cross-sectional study. A lack of unsupportive reactions did not predict reappraisal, which is in line with the findings of Gunzenhauser et al. [9] but not with the retrospective study of Cabecinha-Alati et al. [28], who found an association. However, they also found supportive reactions to be a stronger predictor than unsupportive reactions for reappraisal. Maybe the retrospective view created a recall bias [28]. To foster adaptive ER in children and adolescents it seems to be important to react in a supportive manner with emotion- and problem-focused strategies that serve to validate the emotions of children and focus on the problem.

The second factor for reappraisal in children was reappraisal in parents, which facilitates the use of reappraisal in children. Parents' own ER strategies seem to have had an influence, as proposed by Morris et al. [23]. This result is not in line with the findings of Gunzenhauser et al. [9] who did not find a connection between reappraisal in parents and reappraisal in children. The children explored by Gunzenhauser et al. [9] were very young and mostly preschoolers (mean age 5.11 years). Maybe reappraisal is easier to observe and imitate for older at least school children and the connection becomes stronger with age. To test the influence of age and gender, we explored the influence of age and gender on the parent reported use of reappraisal in an explanatory fashion, but the two factors did not predict reappraisal (see S1 File).

If these results are confirmed, parents' own ER could be an element of future intervention and prevention programs.

There was no negative relation between suppression in parents and reappraisal in children, so this was not in line with the proposition of Gunzenhauser et al. [9] that suppression in parents hinders reappraisal in children. This again might be explained by the age of Gunzenhauser et al.'s sample [9]. In addition, Gunzenhauser et al. [9] explored the families in a longitudinal design and not in a cross-sectional study. It is possible that suppression in the context of parenting behavior is an adaptive strategy [62]. Maybe suppression in parents can be helpful in managing the education process. This might also explain why suppression in parents did not predict the adaptive strategy of reappraisal. Further research to clarify this point is necessary.

Mental health in parents was not associated with the use of reappraisal in children, in contrast with previous results showing an association between depressed mothers and emotion dysregulation development in children about 5 years old [39]. Maybe the influence of mental health in parents on the ER in children decreases with age. Another explanation could be that in the present study, a global score for mental health and not a specific score for depression was used. The data revealed that parents described themselves as quite healthy. This bottom effect might have inhibited an effect of parent's mental health. The SCL-27 does not differentiate very well for samples with low psychopathology symptoms [42]. Thus, future studies should include parents with clinically relevant psychopathology.

Neither parenting behavior, parents' ER, nor mental health in parents predicted the parent reported use of suppression in children. Gunzenhauser et al. [9] found that unsupportive reactions to negative emotions led to suppression in children aged about 5 years. These results could not be replicated and might be explained by the wider age range of our sample. Also, Cabecinha-Alati et al. [28] found this effect in their retrospective study, which was maybe influenced by recall bias [14]. Bariola et al. [14] found the association between parental and child suppression only for mothers. This is in line with Li et al. [31] who found that paternal and maternal ER and parenting behavior and the interaction between the parents contributed differently to the socialization of child ER. The supportive reaction of fathers mediated the relationship between parental emotion dysregulation and father's report of children's ER. In further research, these mechanisms should be examined separately and more differentiated for both parents to understand the different pathways of socialization of ER. Parental mental health did not predict suppression. Maybe again the category mental health was too broad because this relation was especially found for depressed mothers and for very young children (1,5–5 years old) [39]. Changes in the social context from middle childhood to adolescence lead to an increasingly outward orientation [14]. Socializers other than parents (e.g., peers and teachers) could become more important [66] and characteristics of the parents as socializers less important, so age should be considered an important factor. In S1 File, we explored the influence on parent reported suppression of age and gender in an explanatory fashion. Only age predicted the parent reported use of suppression. The use of suppression increased with the age. This was not in line with Gullone et al. [13] who reported a decrease in suppression between 11 and 15 years. But Zimmerman et al. [15] found suppression increasing from 11 years till late adulthood, explaining this with a new organization of ER strategies. The sample explored by Zimmerman et al [15] also included youths older than 15 years like in our sample. Maybe in this period, youths want to seem less vulnerable and show fewer emotions which lead to use more suppression. Further studies should explore the influence of the different ages more closely.

## Limitations and strengths

The present study has some limitations, which should be considered when interpreting the results. First, we relied on questionnaire measures, which might be sensitive to social desirability and response tendencies. Second, we asked parents to report on their own strategies and on the strategies of their children. Results might have been overestimated because of shared source variance. Third, ER strategies also vary with the social context and the interaction partner (e.g. [67]) and have a functionalist component [68]. Further research should also consider using ratings by the adolescents themselves or other people from other contexts, such as teachers, and using a multimethod approach [69]. A multimethod investigation with self-report, psychophysiological data, or a more naturalistic setting, such as an ecological momentary assessment, could be helpful [69]. On the other hand, it might be difficult to create self-report measures for young children [69] and it might be too complex for young children to evaluate their own ER. The advantage of this study is the homogenous data set with a very wide range of ages. Fourth, in the ClinS, the parents knew the therapist and were maybe more ashamed to report their parenting behavior than in the ComS. In the ComS, the survey was conducted completely anonymously. This might have led to social desirability and response biases in parents of the ClinS because they would interact with the person after the survey directly. To control for this effect it would be interesting to assess a waitlist control group that would participate in the survey also online without knowing the future therapist. Otherwise, one advantage of this procedure was the high standard of the diagnostic process because we used qualified therapists, which improved the external validity and we examined a "real" clinical sample, which even looked for therapeutic support. Another limitation might be the heterogeneous diagnoses in the clinical sample. Even if ER seems to be a transdiagnostic underlying factor for psychopathology and there is growing evidence that different diagnoses share common factors [6, 56, 57], there could be specific patterns in ER for different diagnoses. For example, children with anxiety disorders may use other ER patterns than children with anorexia nervosa. In addition, the parenting behavior might differ as mentioned above, depending on the specific diagnoses [27, 59, 60, 61]. On the other hand, the clinical sample is ecological valid with children having comorbid diagnoses as it occurs in "real life" and there is growing evidence for the transdiagnostic approach [6, 56, 57]. Anyway, it might be helpful in further research to analyze the ER patterns and parenting behavior for the different diagnoses in detail to adapt the intervention and prevention programs. A strength of the study was that we explored also both data from fathers and mothers. On the other hand, this might be a limitation because mothers and fathers might influence the socialization process of ER differently. Exploring their data together might create problems to detect these different processes. Further, the groups were not controlled for the income level and education level of parents that are supposed to influence parenting behavior [70]. Maybe this confounded the results and should be controlled for in future studies. A methodological problem was the weak internal consistency for the unsupportive reaction ($\alpha = .53$) of the CCNES-AP, which might have distorted the results. Otherwise, the CCNES with its subscales has been confirmed to be a reliable and valid instrument [54, 71].

Another strength of the study is the homogeneity of the data collected for children and adolescents with a wide range of ages. Previous studies were limited to a smaller range (e.g. [9]. On the other hand, the range of age was wide and we controlled the influence of age and gender only in an explanatory fashion because the sample size was too small to integrate these variables as well. In further research, these factors should be considered more closely. A weakness of the study was the loss of data because of the matching process, but the matching reduced the chance of a bias effect. Because of the matching process, the two groups were comparable

for the two first hypotheses, and age and sex were controlled for. Finally, the study was only a cross-sectional study and we cannot make a statement concerning the direction of the association.

## Implications

Despite the limitations, our findings confirm that reappraisal and suppression are important ER strategies in children and adolescents for maintaining mental health regardless of the diagnosis. This study suggests that emotional dysregulation is an underlying mechanism in a variety of mental disorders and points to support for the transdiagnostic approach [72]. Training in ER strategies can be an important element of prevention and intervention programs [72]. Further, our study confirms that some socializers' characteristics are associated with the use of reappraisal in children. Parents' own use of reappraisal and supportive reactions to negative emotions seems to reinforce the use of reappraisal in children. If future research confirms these results, parents' own use of reappraisal and supportive reactions to negative emotions of children might be helpful and should be promoted. From a clinical perspective, this additional component could be part of prevention and intervention programs for parents, which normally target teaching styles [73].

## Supporting information

**S1 File. Additional analyses concerning age and gender.**
(PDF)

## Author Contributions

**Conceptualization:** Eva-Maria Fassot, Brunna Tuschen-Caffier, Julia Asbrand.

**Data curation:** Eva-Maria Fassot, Julia Asbrand.

**Formal analysis:** Eva-Maria Fassot.

**Supervision:** Brunna Tuschen-Caffier, Julia Asbrand.

**Writing – original draft:** Eva-Maria Fassot.

**Writing – review & editing:** Brunna Tuschen-Caffier, Julia Asbrand.

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
