## [Decision Letter · Decision Letter 0]

6 May 2022

PONE-D-21-33443Association of parental characteristics and emotion regulation in children and adolescents with and without psychopathology: A case control studyPLOS ONE

Dear Dr. Fassot,

Thank you for submitting your manuscript to PLOS ONE. After careful consideration, we feel that it has merit but does not fully meet PLOS ONE’s publication criteria as it currently stands. Therefore, we invite you to submit a revised version of the manuscript that addresses the points raised during the review process.

Based on the advice received, I am willing to consider a revised submission provided it includes major revisions directly addressing the concerns expressed by the reviewers (see below). However, there is no guarantee that a revised manuscript will be accepted for publication. 

When revising your manuscript, please consider all issues mentioned in the reviewers' comments carefully: please outline every change made in response to their comments and provide suitable rebuttals for any comments not addressed.

We look forward to receiving your revised manuscript.

Kind regards,

Claudio Imperatori, Ph.D

Academic Editor

PLOS ONE

2. Please provide additional details regarding participant consent of minors. Specifically, please whether you obtained consent from parents or guardians.

Blank

“I have read the journal's policy and the authors of this manuscript have the following competing interests:The authors declare they have no financial interests. Prof. Tuschen-Caffier is the head of the institute the data was collected.”

Reviewers' comments:

Reviewer's Responses to Questions

**Comments to the Author**

1. Is the manuscript technically sound, and do the data support the conclusions?

Reviewer #1: Partly

Reviewer #2: Yes

2. Has the statistical analysis been performed appropriately and rigorously? 

Reviewer #1: No

Reviewer #2: Yes

3. Have the authors made all data underlying the findings in their manuscript fully available?

Reviewer #1: No

Reviewer #2: No

4. Is the manuscript presented in an intelligible fashion and written in standard English?

Reviewer #1: Yes

Reviewer #2: Yes

5. Review Comments to the Author

Reviewer #1: This in an interesting study reporting an investigation of correlates of reappraisal and suppression in healthy vs fragile children (i.e., with mental health issues). The authors looked for such correlates in parental mental health, parenting behaviors and emotion regulation strategies.

I read this paper with interest and I think that it has potential. Below, I indicate a few suggestions that can help the authors to make a stronger contribution to the literature. Indeed, additional analyses could guide a more focused and robust discussion of the results.

The Introduction is well written and does not need important revision, in my opinion.

First of all, I noticed that recruited children ranged in age from 6 to 18 years old, so from school beginners to almost young adults. Such a wide age range constitutes an advantage if the authors are willing to control for age in the analyses or to make separate models according to age. Otherwise, it might constitute a limitation in itself, as interpretation of results can be biased.

A similar consideration might apply to the composition of the clinical sample. The various diagnoses of children could be grouped in at least two major subgroups (one characterized by more marked difficulties in the internalizing area and the other by more marked difficulties in the externalizing area? just an idea). Otherwise it is difficult to be persuaded that parental variables have a similar effect in regulation strategies of such a heterogeneous sample.

The authors may want to explore a bit more the pattern of relations among parent/child mental health, parenting behaviors and parent/child emotion regulation and try to understand how these variables are associated one to another. For example, Li and colleagues argued that fathers’ regulation might impact on child regulation through supportive behavior (complete ref: Li, D., Li, D., Wu, N., & Wang, Z. (2019). Intergenerational transmission of emotion regulation through parents' reactions to children's negative emotions: Tests of unique, actor, partner, and mediating effects. Children and Youth Services Review, 101, 113-122.)

The authors provided an explanation for results that do not confirm their second hypothesis. However, I think that the authors should also make an effort in explaining why the data did not fully confirm their third hypothesis. Indeed, no significant predictors emerged for suppression. What kind of explanation can be provided? Again, maybe new analyses controlling for age will shed light on this, as the authors reported that Gunzenhauser et al. found that unsupportive parenting behavior led to suppression in young children.

If the authors are going to repeat some of the analyses, another interesting possibility, in my opinion, is to use only mothers’ data. Indeed, many past studies found significant differences between mothers’ and fathers’ variables in affecting children’s outcomes. Therefore, since fathers constitute less than 15% of the sample, the authors could use only mothers’ data (or compare maternal and paternal data). Anyway, lower interest shown by fathers in researches concerning their children is consistently found in the literature and should be acknowledged once again.

Two final points: 1) the authors did not mention gender differences: were there any? 2) the authors reported parental education and income level? was this just to describe the sample or can this info be used in the analyses (for example controlled for?).

Minor points

In my opinion, the use of the acronym “CS” to indicate “community sample” is not effective, as “clinical sample” shares the same initials.. Can the author think of an alternative acronym? Or, even better, two acronyms, one for clinical sample and one for community sample. This would help the reader.

When describing the regression to predict reappraisal, towards the end, the authors write: “The third step including of parents’ mental health remained 334 significant, F(5, 133) = 3.29, p = .008.” However, I find this a bit misleading, as what needs to be highlighted here is that the factor entered in the third step, i.e. parental mental health, was not significant.

Reviewer #2: The current study aimed at assessing any differences in emotion regulation in children with and without psychopathology and whether these difference were affected by their parents' emotion regulation strategies and reaction to their children's negative emotions.

The study is well conducted and technically sound.

However, I have some minor concerns that I would like to see addressed by the authors.

Line 61: there is a missing A for “antecendent”

Line 181: please put the percentage 29% between brackets

Table 2/3: the tables are well done! It seems that there was only a statistical difference for age between the children, but what about the other sociodemographic variables? If there were no differences, then I kindly ask the authors to make it clear in the text, if there were any differences then I kindly ask the authors to add the results in the table and discuss the results in light of the results.

Reaction of parents to the emotions of their children: The authors state that the questionnaire used to assess parents’ reaction to their children’s emotion has six subscales Emotion Focused, Problem Focused, Minimization, Punitive, Expressive Encouragement, and Distress Responses. However, only four seems to have been used: 2 for the supportive reaction (emotion focused and problem focused) and 2 for unsupportive reaction (minimization and punitive). So, I would like to ask the authors about the 2 remaining subscales Expressive encouragement and distress responses. Were they used? If not, why? Because it seems that these two could have been included in the supportive reaction factor and unsupportive reaction factor, respectively.

Results, mental disorders and ER: the authors state that “Children in the CS used the ER reappraisal strategy more often than children in the clinical sample” and “Children in the clinical sample used suppression more often than children in the CS”. However, it was not clear from the materials section that the children completed the ERQ. Could they authors clarify this ambiguity?

Line 330: I believe there is a typo. The table the authors are referring to is Table 5, however, I understood that table was a reference for the MANOVA analysis used to test Hypothesis 2. Maybe the table that the authors wanted to refer to was Table 6. Could the authors please clarify?

Prediction of Children’s ER: since hierarchical regression analyses were used, I kindly ask the authors to add to both Table 6 and Table 7 the R2 and the change in R2

6. PLOS authors have the option to publish the peer review history of their article (what does this mean?). If published, this will include your full peer review and any attached files.

Reviewer #1: **Yes: **Marcella Caputi

Reviewer #2: No

---

## [Author Response · Author response to Decision Letter 0]

19 Jun 2022

Dear reviewers,

thank you very much for reviewing our manuscript. We appreciate your constructive feedback and believe that implementing your ideas and comments has improved our manuscript considerably.

We have highlighted the changes we applied to the manuscript with track changing. Please find our comments on each of your points/ the changes we made below.

Thank you again for your time and effort. 

Reviewer #1

First of all, I noticed that recruited children ranged in age from 6 to 18 years old, from school beginners to almost young adults. Such a wide age range constitutes an advantage if the authors are willing to control for age in the analyses or to make separate models according to age. Otherwise, it might constitute a limitation in itself, as an interpretation of results can be biased. 

Thank you for your suggestion for the supplemental analyses integrating the factors age and gender. As you correctly noted these factors might influence and limit our results. However, the manuscript mainly focuses on relations between parents and children with an additional focus on psychopathology. Thus, the sample size was not adapted to calculate with these additional factors. The power analysis with g*power (ANCOVA with adapted α=.0025, ß.95, f =.25) leads to a necessary sample size of 245 while our sample was limited to 229. Further, the main emphasis of our study was to explore the differences between the community sample and a clinical sample and the influence of parental characteristics. We emphasized this in the introduction, p.6:

 “The present study investigated the differences in ER and parenting behavior between a sample with a vast range of age and heterogenous psychopathology and a community sample.”

We stayed mindful of the length of the manuscript and, thus, focused on some of the mentioned concerns. However, we integrated further points in an online supplement, as these points are really important from another perspective, but are not the main focus of the paper. We referred to these points in the Method section, p.6,7 and the results p.18: 

p.6,7:

“As gender and age of the children might influence ER and parenting behavior, we calculated the different analyzes with gender and age as supplemental variables in an explanatory fashion with the original sample and reported this in our supplement 1. Regarding gender differences and age no specific hypotheses were made because research concerning the influence of age and gender is inconsistent (13-15).

We calculated a MANCOVA with the factors group and gender and the covariate age for hypotheses one and two and integrated age and gender as predictors to the hierarchical linear regression for hypotheses 3. 

Further, we integrated this into the discussion:

p.18:

“The results did not change including age as a covariate and gender as an additional factor (see supplement 1).”

p.18:

“Even the age or gender did not have an influence on the reaction (see supplement 1).”

p.20:

“To test the influence of age and gender, we explored the influence of age and gender on the parent reported use of reappraisal in an explanatory fashion, but the two factors did not predict reappraisal (see supplement 1).”

p.22

“In supplement 1 we explored the influence of age and gender in an explanatory fashion. Only age predicted the parent reported use of suppression. The use of suppression increased with age. This was not in line with Gullone et al. (13) who reported a decrease in suppression between 11 and 15 years. But Zimmerman et al. (15) found suppression increasing from 11 years till late adulthood, explaining this with a new organization of ER strategies. The sample explored by Zimmerman et al (15) also included youths older than 15 years like in our sample. Maybe in this period, youths want to seem less vulnerable and show fewer emotions which lead to use more suppression. Further studies should explore the influence of the different ages more closely.”

“On the other hand the range of age was wide and we controlled the influence of age and gender only in an explanatory fashion because the sample size was too small to integrate these variables as well. In further research, these factors should be considered more closely.”

A similar consideration might apply to the composition of the clinical sample. The various diagnoses of children could be grouped into at least two major subgroups (one characterized by more marked difficulties in the internalizing area and the other by more marked difficulties in the externalizing area? just an idea). Otherwise, it is difficult to be persuaded that parental variables have a similar effect in regulation strategies of such a heterogeneous sample. 

Thank you for your suggestion! We discussed this suggestion extensively. As diagnoses were categorical, and very heterogeneous we decided not to divide the clinical sample. Further, certain diagnoses such as anoxia nervosa or schizophrenia were ambiguous and not easy to group to the internalizing or externalizing area. Another problem was that many children had comorbid diagnoses (29%; e.g., ADHD and an anxiety disorder), so we were not able to divide them into two or more groups, but we integrated this point into the discussion: 

p. 23:

“Another limitation might be the heterogeneous diagnoses in the clinical sample. Even if ER seems to be a transdiagnostic underlying factor for psychopathology and there is growing evidence that different diagnoses share common factors (6, 56, 57), there could be specific patterns in ER for different diagnoses. For example, children with anxiety disorders may use other ER patterns than children with anorexia nervosa. In addition, the parenting behavior might differ as mentioned above, depending on the specific diagnoses (27,59,60,61). On the other hand, the clinical sample is ecological valid with children having comorbid diagnoses as it occurs in “real life” and there is growing evidence for the transdiagnostic approach (6, 56, 57). Anyway, it might be helpful in further research to analyze the ER patterns and parenting behavior for the different diagnoses in detail to adapt the intervention and prevention programs.”

The authors may want to explore a bit more the pattern of relations among parent/child mental health, parenting behaviors, and parent/child emotion regulation and try to understand how these variables are associated one to another. For example, Li and colleagues argued that fathers’ regulation might impact on child regulation through supportive behavior (complete ref: Li, D., Li, D., Wu, N., & Wang, Z. (2019). Intergenerational transmission of emotion regulation through parents' reactions to children's negative emotions: Tests of unique, actor, partner, and mediating effects. Children and Youth Services Review, 101, 113-122.) 

Thank you for this suggestion. We added the results of Li et al. to the introduction p. 5 and the discussion p.22:

 p.5:

“Further, there is some evidence that parents’ ER is associated with their reaction to negative emotions, problems in ER in parents lead to unsupportive behavior (31).”

p.22:

“This is in line with Li et al. (31) who found that paternal and maternal ER and parenting behavior and the interaction between them contributed differently to the socialization of child ER. The supportive reaction of fathers mediated the relationship between parental emotion dysregulation and father’s report of children’s ER. In further research, these mechanisms should be examined separately and more differentiated for both parents to understand the different pathways of socialization of ER.”

The authors provided an explanation for results that do not confirm their second hypothesis. However, I think that the authors should also make an effort in explaining why the data did not fully confirm their third hypothesis. Indeed, no significant predictors emerged for suppression. What kind of explanation can be provided? Again, maybe new analyses controlling for age will shed light on this, as the authors reported that Gunzenhauser et al. found that unsupportive parenting behavior led to suppression in young children. 

Thank you for your suggestion, we tried to explain the results of the third hypothesis more precisely, p.22:

Bariola at al. (14) found the association between parental and child suppression only for mothers. This is in line with Li et al. (31) who found that paternal and maternal ER and parenting behavior and the interaction between them contributed differently to the socialization of child ER. The supportive reaction of fathers mediated the relationship between parental emotion dysregulation and father’s report of children’s ER. In further research, these mechanisms should be examined separately and more differentiated for both parents to understand the different pathways of socialization of ER. Parental mental health did not predict suppression. Maybe again the category mental health was too broad because this relation was especially found for depressed mothers and for very young children (1,5- 5 years old) (39)

If the authors are going to repeat some of the analyses, another interesting possibility, in my opinion, is to use only mothers’ data. Indeed, many past studies found significant differences between mothers’ and fathers’ variables in affecting children’s outcomes. Therefore, since fathers constitute less than 15% of the sample, the authors could use only mothers’ data (or compare maternal and paternal data). Anyway, lower interest shown by fathers in researches concerning their children is consistently found in the literature and should be acknowledged once again. 

Thank you for this suggestion. As the sample size was already small due to the matching process and the ERQ being added later to the survey and taking care of the length and readability of the manuscript, we decided not to calculate the analyses for fathers and mothers separately, but we mentioned this point in the discussion p. 23:

“A strength of the study was that we explored both data from fathers and mothers. On the other hand, this might be a limitation because mothers and fathers might influence the socialization process of ER differently. Exploring their data together might create problems to detect these different processes.”

Two final points: 1) the authors did not mention gender differences: were there any? 2) the authors reported parental education and income level? was this just to describe the sample or can this info be used in the analyses (for example controlled for?). 

As we reported above, we stayed mindful of the length of the manuscript, so we did not address all of the points you mentioned individually but integrated this into the supplement. However, we appreciate your thoughts and while we were aware of the role of gender for the study, your thoughts have extended this perspective. Please let us know if you agree with the section and you’re mentioning or if you’d prefer any changes. Thank you again!

We integrated the statistic for income level and parent education level in tables 1 and 2 and integrated this into the method section p.12 and into the discussion, p.23:

p. 12:

“The difference in age of the children between the two groups was significant, t(227) = -7.37, p = .002. Also, the income level and the education level of the parents differed significantly (see Table 2). To achieve comparability between the two groups, case-control matching was performed with the matching factors age and sex, because these factors are known to influence the ER and parenting behavior (13-15). We decided not to match more variables to keep an acceptable sample size.”

p.23:

 “Further, the groups weren’t controlled for the income level and education level of parents that are supposed to influence parenting behavior (65). Maybe this confounded the results and should be controlled for in future studies.”

In my opinion, the use of the acronym “CS” to indicate “community sample” is not effective, as “clinical sample” shares the same initials.. Can the author think of an alternative acronym? Or, even better, two acronyms, one for clinical sample and one for community sample. This would help the reader.

Thank you for your suggestion, we changed this into ClinS and ComS and hope that this makes the manuscript easier to understand.

When describing the regression to predict reappraisal, towards the end, the authors write: “The third step including of parents’ mental health remained 334 significant, F(5, 133) = 3.29, p = .008.” However, I find this a bit misleading, as what needs to be highlighted here is that the factor entered in the third step, i.e. parental mental health, was not significant. 

We added at p.15: 

“However, no additional variance could be explained (see Table 6).” 

and hope this made it clearer.

Reviewer #2

Line 61: there is a missing A for “antecendent” 

Thank you, we added the “a”

Line 181: please put the percentage 29% between brackets 

Thank you, we corrected this.

Table 2/3: the tables are well done! It seems that there was only a statistical difference for age between the children, but what about the other sociodemographic variables? If there were no differences, then I kindly ask the authors to make it clear in the text, if there were any differences then I kindly ask the authors to add the results in the table and discuss the results in light of the results. 

Thank you for the advice, we integrated the statistic in the table and discussed this at the end: 

p. 23:

“Further the groups were not controlled for income level and education level of parents that are supposed to have an influence on parenting behavior (65). Maybe this confounded the results and should be controlled for in future studies.”

Reaction of parents to the emotions of their children: The authors state that the questionnaire used to assess parents’ reaction to their children’s emotion has six subscales Emotion Focused, Problem Focused, Minimization, Punitive, Expressive Encouragement, and Distress Responses. However, only four seems to have been used: 2 for the supportive reaction (emotion focused and problem focused) and 2 for unsupportive reaction (minimization and punitive). So, I would like to ask the authors about the 2 remaining subscales Expressive encouragement and distress responses. Were they used? If not, why? Because it seems that these two could have been included in the supportive reaction factor and unsupportive reaction factor, respectively. 

Thank you for your suggestion. Different studies using the CCNES did not add these subscales to the scale supportive reaction and unsupportive reaction because the factor analyses by Fabes, Poulin, Eisenberg & Madden-Derdich (2002) revealed that there were only two factors creating the subscales.

The German version of the CCNES by Gunzenhauser, Fäsche, Friedlmeier & von Suchodoletz (2014) (also used only the four subscales like proposed by McElwain, Halberstadt & Volling 2007).

We hope this makes our decision clearer why we used only 4 subscales.

We also added this in the Method section 

p.11: 

A factor analysis by Fabes et al. (53) revealed that there were two factors, supportive and unsupportive reaction to negative emotions, which were created from only four subscales. As recommended by Gunzenhauser et al. (9) for the German version the supportive reaction factor consists of the subscales Emotion Focused and Problem Focused, and the unsupportive reaction factor consists of the subscales Minimization and Punitive.”

Results, mental disorders and ER: the authors state that “Children in the CS used the ER reappraisal strategy more often than children in the clinical sample” and “Children in the clinical sample used suppression more often than children in the CS”. However, it was not clear from the materials section that the children completed the ERQ. Could they authors clarify this ambiguity? 

Because the ERQ was completed only by a part of the sample (children older than 11) we used the parent reported ERQ. We tried to clarify this in the manuscript, p.14: 

“Children in the ComS were reported by their parents to use the ER reappraisal strategy more often than children in the ClinS, t(56) = -2.39, p = .020. Children in the ClinS were reported by their parents to use suppression more often than children in the ComS, t(56) = 2.79, p = .007.”

We hope that this made it clearer.

Line 330: I believe there is a typo. The table the authors are referring to is Table 5, however, I understood that table was a reference for the MANOVA analysis used to test Hypothesis 2. Maybe the table that the authors wanted to refer to was Table 6. Could the authors please clarify? 

Thank you, we changed this into table 6.

Prediction of Children’s ER: since hierarchical regression analyses were used, I kindly ask the authors to add to both Table 6 and Table 7 the R2 and the change in R2

Thank you for the advice. We added R² and change in R² to table 6 and table 7.

---

## [Decision Letter · Decision Letter 1]

5 Jul 2022

Association of parental characteristics and emotion regulation in children and adolescents with and without psychopathology: A case-control study

PONE-D-21-33443R1

Dear Dr. Fassot,

We’re pleased to inform you that your manuscript has been judged scientifically suitable for publication and will be formally accepted for publication once it meets all outstanding technical requirements.

Kind regards,

Claudio Imperatori, Ph.D

Academic Editor

PLOS ONE

Additional Editor Comments (optional):

Reviewers' comments:

Reviewer's Responses to Questions

**Comments to the Author**

1. If the authors have adequately addressed your comments raised in a previous round of review and you feel that this manuscript is now acceptable for publication, you may indicate that here to bypass the “Comments to the Author” section, enter your conflict of interest statement in the “Confidential to Editor” section, and submit your "Accept" recommendation.

Reviewer #1: All comments have been addressed

Reviewer #2: All comments have been addressed

2. Is the manuscript technically sound, and do the data support the conclusions?

Reviewer #1: (No Response)

Reviewer #2: Yes

3. Has the statistical analysis been performed appropriately and rigorously? 

Reviewer #1: (No Response)

Reviewer #2: Yes

4. Have the authors made all data underlying the findings in their manuscript fully available?

Reviewer #1: (No Response)

Reviewer #2: No

5. Is the manuscript presented in an intelligible fashion and written in standard English?

Reviewer #1: (No Response)

Reviewer #2: Yes

6. Review Comments to the Author

Reviewer #1: (No Response)

Reviewer #2: I thank the authors for addressing all my comments.

I have endorsed the publication of the manuscript.

7. PLOS authors have the option to publish the peer review history of their article (what does this mean?). If published, this will include your full peer review and any attached files.

Reviewer #1: **Yes: **Marcella Caputi

Reviewer #2: No

---

## [Editor Report · Acceptance letter]

7 Jul 2022

PONE-D-21-33443R1 

Association of parental characteristics and emotion regulation in children and adolescents with and without psychopathology: A case-control study 

Dear Dr. Fassot:

I'm pleased to inform you that your manuscript has been deemed suitable for publication in PLOS ONE. Congratulations! Your manuscript is now with our production department. 

Kind regards, 

on behalf of

Dr. Claudio Imperatori 

Academic Editor

PLOS ONE